# Silver Nanostructures: Limited Sensitivity of Detection, Toxicity and Anti-Inflammation Effects

**DOI:** 10.3390/ijms22189928

**Published:** 2021-09-14

**Authors:** Olga V. Morozova

**Affiliations:** 1Federal Research and Clinical Center of Physical-Chemical Medicine, Federal Medical Biological Agency, Malaya Pirogovskaya Street 1a, 119435 Moscow, Russia; morozova2010@gmail.com; 2Ivanovsky Institute of Virology, National Research Center of Epidemiology and Microbiology of N. F. Gamaleya, Russian Ministry of Health, Gamaleya Street 16, 123098 Moscow, Russia

**Keywords:** silver ions, nanopaticles, nanoconjugates with silver core and protein shells, stability, cytotoxicity, immunosupression

## Abstract

Nanosilver with sizes 1–100 nm at least in one dimension is widely used due to physicochemical, anti-inflammatory, anti-angiogenesis, antiplatelet, antifungal, anticancer, antibacterial, and antiviral properties. Three modes of the nanosilver action were suggested: “Trojan horse”, inductive, and quantum mechanical. The Ag^+^ cations have an affinity to thiol, amino, phosphate, and carboxyl groups. Multiple mechanisms of action towards proteins, DNA, and membranes reduce a risk of pathogen resistance but inevitably cause toxicity for cells and organisms. Silver nanoparticles (AgNP) are known to generate two reactive oxygen species (ROS)-superoxide (•O_2_^−^) and hydroxyl (•OH) radicals, which inhibit the cellular antioxidant enzymes (superoxide dismutase, catalase, and glutathione peroxidase) and cause mechanical damage of membranes. Ag^+^ release and replacement by electrolyte ions with potential formation of insoluble AgCl result in NP instability and interactions of heavy metals with nucleic acids and proteins. Protein shells protect AgNP core from oxidation, dissolution, and aggregation, and provide specific interactions with ligands. These nanoconjugates can be used for immunoassays and diagnostics, but the sensitivity is limited at 10 pg and specificity is restricted by binding with protective proteins (immunoglobulins, fibrinogen, albumin, and others). Thus, broad implementation of Ag nanostructures revealed limitations such as instability; binding with major blood proteins; damage of proteins, nucleic acids, and membranes; and immunosuppression of the majority of cytokines.

## 1. Introduction

Nanosilver is a generic term that refers to nanoscale Ag materials that have at least one dimension less than 100 nm, and which are commonly in the form of particles called silver nanoparticles (AgNP). They remain the most used engineered nanostructures in approximately ~500 commercialized consumer products with annual production 320 tons [1]). Due to physicochemical properties such as unique light propagation, enhanced reactivity of hot electrons, and the relaxation of excited NP, AgNP are used in nanomedical devices as tools for medical diagnostic imaging and biosensing [2]. Because nanosilver induces numerous oxidative and non-oxidative processes including adhesion to the surfaces, electrostatical interaction with membranes, cell destruction and inappropriate functioning of organelles, and interaction with protein and nucleic acids, AgNP are also employed as antifungal, antibacterial, and antiviral drugs [3,4] for wound dressings, long-term burn care products [5], medical device coating, medical textiles, orthopedics, contraceptive devices, cosmetic clothing and lotions for both treatment and supplementary drug and/or nutrient delivery, paints, sunscreens, etc. [2,6,7]. Although other metals such as copper (Cu), zinc (Zn), iron (Fe), lead (Pb), aluminum (Al), and gold (Au) are also effective for disinfection, silver (Ag) remains the most efficient and therefore the best studied but moderately understood since ancient times [8] until the ongoing COVID-19 pandemic [9]. Besides this, Ag-containing nanostructures are currently used in electronics, the food industry, and in households [6,7].

There are physical, chemical, physicochemical, and biological methods of nanosilver fabrication. Physical methods include thermal evaporation or condensation, laser ablation, arc discharge, metal sputtering and lithography. Chemical methods are citrate method, borohydrate reduction, tollens (silver mirror) reaction, and polyol process. Physicochemical approaches are photoinduced reduction, reduction induced by plasma or by laser irradiation, discharge-induced reduction, thermal decomposition, and electrochemical approaches. The most numerous methods are so-called biological methods by using amino acids, peptides, proteins, nucleic acids, polysaccharides, flavonoids, terpenes, and oils, as well as plant or fungi extracts, whole bacteria, yeast cells, and viruses. Biological approaches are found to be simple, environmental, commercial, and single-step methods that do not require elevated temperature, pressure, force, and chemicals [7]. Recovery of Ag^+^ by using various plant extracts, bacteria, biodegradable polymers, enzymes, and even microwaves [10]—is gradually replacing harmful chemical synthesis and energy-consuming physical approaches of the nanosilver fabrication [7,10,11,12].

Inevitably, from the rapid growth in its manufacture and utilization follows an increased environmental and human exposure. Despite the evergrowing presence of the nanosilver in the market and extensive research on potential acute and chronic toxicity in vitro and in vivo, exact multiple mechanisms remain uncertain. Remarkably high surface-to-volume ratio of the nanosilver enhances surface reaction properties, thus increasing the possible interaction with biological fluids including blood serum, lymph, saliva, mucus, or lung lining fluid components. AgNP sizes, shape, and surface coating contribute to interactions with biomaterials, organic and inorganic molecules, cells, and viruses [13].

Evident concerns regarding the overuse of the nanosilver with possible interactions with biological systems in new unpredicted ways [14] and the need for a risk–benefit analysis for all applications and eventually restrictions remain emerging [13]. Moreover, insufficient data are currently available about the principal restrictions for the nanosilver to use as a diagnostic and therapeutic agent. The American Conference of Governmental Industrial Hygienists has established threshold limit values for metallic silver (0.1 mg/m^3^) and soluble compounds of silver (0.01 mg/m^3^) [15]. Meanwhile European Commission Scientific Committee on Emerging and Newly Identified Health Risks (SCENIHR) concluded from available published studies that AgNP might have different toxicological properties from the bulk substances and ions, but their risks should be assessed on a case-by-case basis [14].

Current research is aimed at stability, toxicity, and immunomodulation potential of the nanosilver.

## 2. Stability of Ag Ions, Citrate-Coated AgNP, and Their Nanoconjugates with Proteins

Both microfluid diagnostics and treatment of diseases require maintenance of the nanosilver concentrations [6]. AgNP exhibit a “core−shell structure” with metallic silver in their central core, surrounded by the surface oxide or sulfide layers as the outer shell [16]. Ag^+^ ions release in the process of AgNP dissolving. Replacement of Ag^+^ by electrolyte ions, potential formation of insoluble AgCl, subsequent catalyzed oxidative corrosion of Ag, and further dissolution of surface layer of Ag_2_O take place [16,17]. Oxidative dissolution of metallic AgNP in the presence of an electron acceptor is catalyzed by nucleophilic reagents that change the chemical potential or Fermi level at the particle surface. This oxidation is controlled by the difference in the chemical potential between AgNP (with nucleophilic or stabilizing agents) and an electron acceptor. For uncoated silver nanoparticles, the oxidation of Ag(0) to Ag^+^ at the particle surface shifts the chemical potential of the particle to a more positive value, and if it approaches that of the electron acceptor (e.g., O_2_), oxidation ceases. The opposite shift in the potential occurs for metallic AgNP with adsorbed nucleophiles (e.g., Cl^−^ or NO_3_^−^), resulting in an increase in the oxidation of Ag(0) [16]. Therefore, AgNP in the presence of ions and especially after addition of EDTA are not stable due to oxidation, dissolution, and aggregation during a few hours. Nanosilver is evolving or ageing in contrast to dissolved Ag species [13]. However, our measurements using UV-visible spectroscopy, dynamic light scattering (DLS), and scanning electron microscopy (SEM) revealed that the citrate-coated AgNP remained stable colloid solutions in deionized water at room temperature for decades but not in the presence of ions. Citrate-coated AgNP are not stable in the presence of phosphate buffer solution (PBS) (0.01 M Na_2_HPO_4_/KH_2_PO_4_, 0.15 M NaCl/KCl) during 1 h at room temperature (Figure 1) due to replacement of Ag^+^ by Na^+^ and K^+^ ions [16,17]. To prevent AgNP dissolution and aggregation, various surfactants and polymers are introduced during or after synthesis [16]. Coating layers may enhance electrostatic and steric repulsion. Adsorption of polymers or nonionic surfactants provides steric hindrances depending upon the thickness of the adsorbed layer [16]. Recovery of Ag^+^ by using various plant extracts, bacteria, biodegradable polymers, enzymes, and even microwaves is gradually replacing harmful chemical synthesis and energy-consuming physical approaches of the nanosilver fabrication [11,12,18].

Nanosilver, like other nanostructures immediately after administration into an organism, becomes wrapped by serum and cellular proteins forming protein corona. These natural spontaneous protein shells decrease the efficiency of targeting by directing nanostructures to the reticuloendothelial system by masking the active targeting moieties and decreasing their ability to bind to their target receptor but may re-direct NP towards endogenous targets. Surface of the nanosilver dynamically adsorbs proteins, forming a robust rapidly exchanging “biocorona”. A hard corona with long-term stability can be formed with immunoglobulins IgG/IgM and fibrinogens and may alter NP size, shape, surface charge, and agglomeration state, as well as cellular toxicity and internalization, trafficking, opsonization, and eventually pattern of biodistribution [17]. Colloids of Ag possess high affinity for binding with serum albumins; their ability to bind with *Staphylococcus aureus* protein A is less efficient, whereas a number of proteins (for example, human immunodeficiency virus (HIV-1) envelope antigen) cannot attach to AgNP at all. Despite known chemical affinity of sulfur atoms to precious metals, direct correlation between cystine disulfide bridge content and binding with AgNP was not observed, perhaps because of strong bonds between two cysteines that stabilize protein conformations.

The NP stability depends on the affinity of coating molecules to the particle surface; repulsion from neighboring molecules; loss of chain entropy upon adsorption; and also nonspecific dipole interactions between the macromolecule, the solvent, and the surface. Protein coronaprotects AgNP from dissolution and aggregation (Figure 1). The nanoconjugates of the noble metal NP with proteins remain stable at +4 °C for several months [17].

Consequently, AgNP should not be used in microfluidic diagnostics and treatment because of their dissolution and aggregation in electrolyte solutions in any bio-relevant physiological media. Preliminary binding of AgNP with some proteins permits the construction of soft corona, which dynamically exchange with major protective proteins such as immunoglobulins, fibrinogen, and albumins immediately after administration in blood sera or other biological fluids. Different affinity of AgNP binding with proteins determines stability and specificity of the nanoconjugates.

## 3. Nanosilver in Diagnostics

Physicochemical features of the nanosilver with unique plasmon-resonance optical scattering properties determine possible implementation in diagnostics [6,19]). High surface-to-volume ratio and affinity to phosphate, carboxyl, amino, and thiol groups provide interaction of the nanosilver with DNA, proteins, polysaccharides, and phospholipides [12,20]. The resulting nanoconjugates may be applied in immune and genodiagnostics in biosensors with microchips [19] for bio-imaging using transmission electron microscopy. However, Ag^+^ ions may intercalate between purine and pyramidine base pairs and disrupt the hydrogen bonding between the two anti-parallel strands, thus denaturing the template DNA molecules [8,21] and disturbing possible molecular hybridization with labeled probes. Binding of AgNP with proteins is weaker compared to AuNP, but protein corona can be formed with the majority of proteins including the main blood proteins [17]. However, close proximity of surface biopolymers to AgNP and possible leakage of Ag^+^ cations deteriorate their conformations and inhibit surface ligands. Extinction, light scattering, surface plasmon resonance (SPR), and surface-enhanced Raman spectroscopy (SERS) of AgNP exceed those of AuNP in 10–100 times. Relatively low price is also an advantage of the nanosilver.

The stable nanoconjugates of AgNP with immunoglobulins of different origin, classes, and specificity, including both polyclonal and monoclonal antibodies, were constructed by (1) direct binding of AgNP with purified IgG or IgM [17], (2) nanoprecipitation of proteins from their solutions in fluoroalcohols [22], (3) physisorption of proteins on the AgNP surface treated with poly(allylamine)s, and (4) encapsulation of AgNP into SiO_2_ envelopes with subsequent functionalization with organosilanes. Physisorption of proteins on surfaces of AgNP is reversible, and up to 70% of the attached proteins can be eluted. AgNP possess high affinity for binding with immunoglobulins but not with any protein. SiO_2_ layer on surfaces of metal NP is suitable for silanization and covalent attachment of any protein. The developed methods of fabrication of AgNP with protein shells permit attaching any protein at different distances from the metal core to avoid possible inactivation of proteins, to reduce fluorescence fading, and to stabilize the nanoconjugates [17]. Green synthesis with plant extract permits the construction of shells consisting of tannin, flavonoids, terpenoids, saponins, steroids, phenolics, carbohydrates, anthraquinones, and cardiac steroidal glycosides for the nanosilver stabilization and their possible intracellular delivery [23].

To detect binding of immobilized antigens with AgNP conjugated with IgG, the analyzer based on light scattering of dark field laser of total internal reflection with the wavelength 532 nm and corresponding software were used. The sensitivity limit of the nanosilver-based immunodiagnostic systems was nearly 10 pg/dot for direct binding of AgNP with immobilized IgG and 100 pg for 3-layer sandwich immunoassay. For comparison, thresholds of ELISA and xMAP multiplex immunofluorescent analysis with fluorescent magnetic microspheres were 1 pg/mL. Specificity of Ag nanoconjugates is limited due to their binding with the major blood serum proteins: IgM, IgG, fibrinogen, and albumins with high background level. Protein dots on NH_2_^−^ and COOH^−^ modified surfaces of chips are not homogenous, causing problems of dot-to-dot reproducibility.

Taken together, immunodiagnostics based on AgNP covered with IgG shells remains inferior compared to specificity and sensitivity of the widely used ELISA and xMAP for 10 to 100 times. Specificity of immunodetection and ratio of signal to background are limited because of AgNP binding with blood proteins.

## 4. Toxicity of Nanosilver

Multiple mechanisms of nanosilver toxicity remain uncertain, despite three recently suggested modes: “Trojan horse”, inductive, and quantum mechanical, which may have a collaborative effect [12].

### 4.1. “Trojan Horse” Mechanism

Silver in ionic, nanoparticulate, and bulk forms behave very differently. Due to their large surface area, AgNP are able for rapid oxidation, dissolution, reactive capacity, and binding with inorganic and organic molecules [15,16]. When the size of metallic silver is shrunk to a nanometer scale, it can enter the cells and cause adverse effects [15]. Nanosilver enters into cells either by endocytosis or by diffusion.

AgNP can serve as Ag^+^ carriers with release of Ag^+^ ions through the oxidative dissolution [13,16,17]. The Ag^+^ cations have an affinity to amino, thiol, phosphate, and carboxyl groups. Therefore, silver-containing complexes can interact with proteins, DNA, polysaccharides, and phospholipides. Consecutive steps of cytotoxicity include adhesion to cell surface, altering the membrane properties; reactive oxygen species (ROS) formation, damaging cellular, bacterial membranes, and viral envelopes; interactions with DNA; and enzyme deterioration [24]. Besides oxidative stress induction, heavy metal ion release that occurs in aqueous solutions of electrolytes, producing biologically active Ag^+^ [25,26] and non-oxidative mechanisms, were suggested for silver nanostructures [27]. The ROS generation inhibits the antioxidant defense system and causes mechanical disruptions of the viral envelopes and cellular membranes. Metal ions are slowly released from metal oxide and are absorbed through the cell membranes or viral envelopes, followed by direct interaction with the functional groups of proteins and nucleic acids, such as mercapto (−SH), amino (−NH_2_), and carboxyl (−COOH) groups, damaging enzyme activity, changing their structure, affecting the normal physiological processes, and ultimately inhibiting the pathogens of different origin [14,28]. Additional mechanisms of Ag^+^ antimicrobial action are becoming evident. Ag^+^ ions may react with phosphorus and sulfur groups of surface proteins after posttranslational modification (phosphorylation and sulfation) on cellular membranes, bacterial cell walls, and virions. Ag^+^ cations electrostatically interact with membrane phospholipids and proteins, which can cause the depolarization and destabilization of cellular membranes, viral envelopes, and leakage of H^+^. Thus, the nanosilver increases membrane permeability, reduces mitochondrial membrane potential, and inactivates bacterial or mitochondrial inner membrane-associated respiratory chain including cytochromes. Mitochondria appear to be sensitive targets for the nanosilver toxicity [12]. Dose-dependent effects of silver ions on cell replication and other developmental endpoints in mammalian cells indicate that Ag^+^ cations alter mitochondrial functions, resulting in the release of apoptogenic signals and subsequent cell death [29].

AgNP induce ROS and release of cytochrome C into the cytosol and translocation of Bax protein to mitochondria [6]. Increased ROS along with significant depletion of the antioxidant enzymes and glutathione result in oxidative stress. Oxidative stress occurs when generation of ROS exceed the capacity of the antioxidant defense mechanism. Depletion of glutathione and protein-bound sulfhydryl groups and damage in the activity of various antioxidant enzymes indicative of lipid peroxidation have been implicated in oxidative damage. ROS and oxidative stress interrupt ATP synthesis, implicate the mitochondria-dependent jun-*n*-terminal kinase, and cause DNA damage and apoptosis. Interaction of AgNP with DNA hinders DNA replication and leads to cell cycle arrest at the G1, G2/M phase and complete blockage in S phase [6,27]. Little is currently known as to how the nanosilver binds and destroys RNA [6]. Nanosilver prevents translation of protein due to damage of ribosomal 30S subunits. However, the impact of metal ions on the pH inside lipid vesicles is small and has weak antimicrobial activity. Therefore, dissolved metal ions are not determined as the main antimicrobial mechanism of AgNP. Moreover, heavy metal ions can indirectly act as carriers of antimicrobial substances [27]. Thus, disruption of cellular membranes and viral envelopes and interactions with DNA and proteins [26,27] are the major known processes of silver-induced disinfecting activity. The numerous mechanisms of action against infectious agents would require multiple simultaneous gene mutations for resistance to develop; therefore, a resistance to silver-containing compounds and nanostructures is hardly possible [27].

Noteworthy, that AgNP cytotoxicity exceeds that of Ag+. For instance, cytotoxic concentration of AgNO_3_ for 50% of human larynx carcinoma HEp-2 cells (CC_50_) was 50 μg/mL, whereas for the nanoconjugates of AgNP with protein shells, CC_50_ was 1.4 μg/mL (calculations are based on the calibration curve of atomic absorption spectroscopy (AAS) data). At present there is no evidence of the efficient uptake or intracellular localization of the citrate-coated AgNP conjugated with fluorescent proteins. Intracellular Ag^+^ release appears to be responsible for the toxicity since the cultural media after treatment of cells with the nanosilver do not cause any cytotoxicity. Thus, AgNP cytotoxicity is mainly associated with the rate of intracellular Ag release, a “Trojan horse” effect [1,12,13,27].

### 4.2. Inductive Toxicity Mechanism

Ag^+^ ions electrostatically interact with membrane phospholipids and proteins, which can cause the depolarization and destabilization of cellular membrane and leakage of H^+^. AgNP induce disorder at the molecular level of lipid bilayers, making them more fluidic and expanded. AgNP adhesion can occur through electrostatic attraction or weak interaction forces [6]. Moreover, AgNP themselves can disorder the different cell organelles function due to adhesion to their surface. Thus, the nanosilver can induce changes in structure or activate cellular destructive mechanisms causing organelles’ inappropriate functioning. Therefore, it was suggested to call “inductive” toxicity mechanism [12].

### 4.3. Quantum Mechanical Mechanism

Plasmon modes and quantum states of the metallic NPs are the unique light propagation, enhanced reactivity of hot electrons, or the relaxation mechanisms of excited NP. The hot electrons in noble NP can decay through two different actions, namely, electron–electron and electron–phonon interactions. The successful transfer of Ag NP energy, provided by surface plasmon resonance (SPR) through electron–electron interactions requires the direct substrate molecule deposition on the NP surface. The ROS formation under the irradiation was accompanied by the decrease in the superoxide dismutase, catalase, and glutathione peroxidase activity. In contrast, the electron–photon interactions lead to thermal diffusion (heat transfer) outside the NP, which is the basis for photothermal therapy. The NP are more stable under irradiation, and metallic NP have significantly higher extinction coefficient due to SPR, which additionally can be tuned by changes in NP forms and sizes. Ag NP size has a significant impact on their features, which is related to the quantum size effect [12].

Besides short-term or acute toxicity described above and observed within several days after addition of the nanosilver to eukaryotic tissue cultures, bacterial suspensions, and viruses in in vitro experiments, one should not exclude long-term or chronic toxicity [6]. AgNP have the potential to induce genes associated with cell cycle progression, DNA damage, and apoptosis at non-cytotoxic doses after long exposure [6].

AgNP have been reported to be toxic to human cell lines [30]. Cellular uptake of AgNP takes place either via diffusion (translocation), endocytosis, or phagocytosis. Upon entering the cytoplasm, AgNP themselves or Ag^+^ ions can generate ROS, leading to DNA damage, protein denaturation, and apoptosis. AgNP of different sizes and shapes tend to accumulate in the mitochondria, thereby inducing mitochondrial dysfunction, i.e., a reduction in mitochondrial membrane potential, and promoting ROS creation [30]. AgNP cytotoxicity in mammalian cells depends on the NP sizes, shape, surface charge, dosage, oxidation state, agglomeration condition, and cell type. They induce a dose-, size-, and time-dependent cytotoxicity, particularly for NP with sizes less 10 nm.

Surface charge of AgNP stabilized with citrate anions or protein envelopes is a parameter responsible for cellular uptake. In particular, high-level toxicity of positively charged nanoconjugates versus negatively charged coatings has been reported. It can be caused by the adhesion of AgNP onto the negatively charged cell membranes, their consequent entry to the cell, potential release of Ag^+^ inside the cell, damage of cellular proteins and nucleic acids, and other cytotoxic effects. For instance, the following coatings posses surface charges: (1) positive: polyethylenimine, chitosan, poly-L-lysine, and cetyltrimethylammonium bromide; (2) negative: bovine serum albumin (BSA), citrate, odium bis(2-ethylhexyl)-sulfosuccinate; (3) neutral: polyvinylpyrrolidone. Ag^+^ ions alter mitochondrial function, resulting in the release of apoptogenic signals and subsequent cell death. They may destroy DNA-dependent DNA replication, RNA transcription, and translation.

Thus, in vitro AgNP are toxic to a variety of examined tissues including lung, liver, brain, vascular system, and reproductive organs. AgNP concentrations used in these studies were 1–200 μg/mL, with most in the 5–50 μg/mL range. The data coincide with our calculations of CC_50_ for three human cell lines in a range. However, expression induction of genes involved in cell cycle and apoptosis was noticeable at <1 μg/mL of the nanosilver [6]. Moreover, the experimentally observed cytotoxic concentrations significantly exceed limits recommended by the American Conference of Governmental Industrial Hygienists (0.01 mg/m^3^ for any soluble compounds of silver corresponding to 10 pg/mL) in order to reduce a risk of chronic toxicity [15].

The common mass-only dose metric model employed in toxicology for traditional substances is evidently not convenient for engineered nanomaterials. Alternative dose metrics include particle number, ion release (kinetics, equilibrium), and the total particle surface area. Nevertheless, polydisperse particle suspensions, the ambiguity in the surface area, and concentrations will obscure the analysis. Therefore, the Organization for Economic Cooperation and Development recommended that particle number, surface area, and mass should all be measured when possible, in order to enable calculation of alternative dose metrics. For AgNP, both surface area and ion release have been reported as effective alternative dose metrics for nanotoxicological assessment. Nanosilver-mediated cytotoxicity in mammalian cells depends greatly on the nanoparticle size, shape, surface charge, dosage, oxidation state, and agglomeration condition, as well as the cell type. Smaller AgNP cause more toxicity than larger ones, owing to their larger surface area and reactivity [30].

Studies in vitro indicated that AgNP are toxic to the mammalian cells derived from skin, liver, lung, brain, vascular system, and reproductive organs. They suggested multiple targeting of the nanosilver in organisms. Bio-distribution and toxicity studies in vivo confirmed that AgNP administered by inhalation, ingestion, or intraperitoneal injections were subsequently detected in blood and damaged several organs, including the brain [6]. Moreover, AgNP exerted developmental and structural malformations in non-mammalian model organisms typically used to elucidate human disease and developmental abnormalities [6].

Long exposure of humans to the nanosilver from cations to NP that can penetrate in living organisms via several routes including inhalation, oral ingestion, intravenous injection, and dermal contacts can lead to argyria, or skin discoloration, and argyrosis, or discoloration of the eyes, as soluble silver incorporates into the tissues with permanent damage of skin microvessels and eyes [30]. Studies in vivo showed AgNP accumulation in liver, spleen, and lungs of experimental animals. However, currently available data about toxicity of silver nanowires (AgNW) (micron-range long with diameters <100 nm) remain contradictory (Liao et al., 2019). For both short (1.5 mm) and long (10 mm) AgNW, after inhalation lung inflammation at day 1, disappearance by day 21 has been described, and in bronchoalveolar lavage fluid, long AgNW cause neutrophilic and macrophagic inflammation [29].

Exposure to different forms of the nanosilver leads to distinct outcomes. Whereas elemental silver is not associated with health effects, soluble Ag is associated with lowered blood pressure, diarrhea, respiratory irritation, and fatty degeneration in the liver and kidneys. Furthermore, after different routes of administration including intravenous, intraperitoneal, and intratracheal ways, AgNP can cross the brain–blood barrier in vivo and tends to accumulate in the liver, spleen, kidney, and brain [15]. Respiratory tract, gastrointestinal tract, skin, and female genital tract are the main entry portals of the nanosilver into the human body through direct substance exchange with the environment. Additionally, systemic administration is also a potential route of entry, since colloidal AgNP have been exploited for diagnostic imaging or therapeutic purposes. Inhalation and instillation experiments in rats showed that low concentration, but detectable, ultrafine silver (14.6 ± 1.0 nm) initially appeared in the lung and was subsequently distributed to the blood and other organs, such as the heart, liver, kidney, and even brain. Nanosilver accumulates in the blood, liver, lungs, kidneys, stomach, testes, and brain. AgNP less than 12 nm affect early development of fish embryos and cause chromosomal aberrations and DNA damage.

Animal and human studies indicate that it is difficult to remove silver completely once it has been deposited in the body; however, the nanosilver can be excreted through the hair, urine, and feces. Moreover, liver cells may develop a metabolic-based protection against AgNP and Ag^+^.

The nanosilver penetration through the blood–brain barrier is still debatable. Inhaled AgNP may reach the brain through the nasopharyngeal system. They deposit in the olfactory mucosa of the nasopharyngeal region and subsequently are translocated into the brain via the olfactory nerve [6]. Additional evidence is dopamine depletion in neuroendocrine cells [6]. However, even in the absence of Ag in cerebrospinal fluid, Ag-mediated neurotoxic complications such as hypoactivity or reverse increased vivacity, as well as changes in noradrenaline, dopamine, and 5-HT concentrations in the brain, were observed. Upon oral exposure to AgNO_3_, the main target organs include the liver and spleen, followed by the testes, kidney, brain, and lungs; AgNP are formed in vivo from Ag+ ions and they are probably composed of silver salts. The elimination of silver from brain and testes is extremely slow [29]. AgNP may translocate into the central nervous system through damaged blood–brain barrier, nerve afferent signaling and eye-to-brain ways, and even through olfactory receptors of the brain neurons. AgNP could stimulate the activation of glial cells to release proinflammatory cytokines and generate ROS and nitric oxide production, resulting in the neuroinflammation, including several immune response-relevant signaling pathways [31]. While Ag_2_S deposits have been seen in the region of cutaneous nerves, there is no indication of toxic risk of silver to the peripheral nervous system [29].

Nanosilver is able to access the human reproductive system via a variety of commercial products such as contraceptive devices and feminine hygiene products. AgNP cause toxicity to germ line stem cells via reduction in mitochondrial function, induction of membrane leakage, and apoptosis. Examination of the DNA damage response to nanosilver in mammalian embryonic stem cells and embryonic fibroblasts showed apoptosis via upregulation of cell cycle checkpoint protein p53, DNA damage repair protein Rad51, and phosphorylated-H2AX [6]. Polysaccharide-coated silver NP are more individually distributed, while agglomeration of the uncoated particles limits the surface area availability and access to membrane-bound organelles. Therefore, AgNP coated with starch, bovine serum albumin (BSA), or polyvinyl pyrolidine (PVP) induced more severe damage versus uncoated nanosilver [6].

Endocytosis and exocytosis of AgNP occur simultaneously and depend on physicochemical properties of NP and protein corona [32]. All nanostructures preferentially accumulate in tumor cells due to the enhanced permeability and retention (EPR) effect. The tumors possess a leaky vasculature and lack lymphatic drainage, allowing the AgNP to reside at the tumor site for a longer duration compared to the free drug molecules [32]. Cytotoxicity of the nanosilver for tumor cells provides anticancer properties.

Evergrowing production of nanosilver increases their release into aquatic environments. Once AgNP reach fresh water, the sea, or underground water, they oxidize into Ag^+^ ions that are toxic to aquatic organisms. Later on, Ag^+^ cations can bind with abundant anions available in the environment with formation of sparingly soluble salts AgCl or Ag_2_S. Marine inhabitants (shrimps, prawns, crabs, lobsters, and crayfish) are known to be much more vulnerable to the impacts of silver than bacteria [29]. By accumulating in aquatic organisms, AgNP can enter the human body.

Taken together, the distribution of AgNP after their inhalation, ingestion, and oral and subcutaneous administration involves a variety of target organs including the lungs, liver, and brain. The brain appears to be the most sensitive. The nanosilver causes expression of genes associated with cell cycle progression and related to apoptosis. Therefore, Ag nanostructures remain unsafe, even at non-cytotoxic doses less than 0.5 μg/mL [6].

## 5. Protection Mechanisms of Cellular and Host Defense against Nanosilver

Various cellular defense mechanisms, innate immunity of vertebrates, and accumulation in certain organs for metabolic-based degradation and subsequent elimination of the nanosilver provide relative protection.

The formation of protein corona on NP surfaces upon the in vivo administration is inevitable. Protein shells of all nanostructures after their systemic administration may explain the lack of the in vitro–in vivo correlations and the preclinical to clinical extrapolation. Protein shells provide stability of the nanocongugates, decrease their cytotoxicity, and determine the interaction of NP with the target and non-target cells. The chemical composition of protein corona may serve as a fingerprint for NP of certain types since different NP tend to recruit cellular and serum proteins to variable extents. Vitronectin mediates accumulation in integrin receptor-expressing melanoma cells both in vitro and in vivo, while complement 3 protein (C3), an opsonin and dysopsonin regulate the balance between the reticuloendothelial system uptake and blood circulation [32].

The cytotoxic effect and oxidative stress of Ag^+^ ions on mouse lung macrophage cells result in necrotic rather than apoptotic cell death by reducing functional sulfhydryl groups in the cells [29].

Another defense mechanism performed by the most abundant leukocytes, polymorphonuclear neutrophils (PMN), is based on neutrophil extracellular traps (NETs) [33]. NETs are extracellular fibers consisting of DNA with histones (H3), myeloperoxidase, and neutrophil elastase. NETs form a barrier that hinders the transmission of pathogens and due to high local concentrations of antimicrobial proteins degrade virulence factors and kill bacteria. However, high concentrations of active proteins may cause host immune damage, contribute to platelet aggregation, and cause thrombosis [33,34]. The nanosilver activates polymorphonuclear neutrophils to release NETs but does not alter the extracellular lactate dehydrogenase level [34].

Nanosilver toxicity results in host defense, including apoptosis, necrosis [2], and NETs formation [34]. Moreover, AgNP, but not Ag^+^ ions, decrease the viability and the cytotoxic potential of natural killer (NK) cells secreting cytokines and killing damaged cells and amplified the expression of the inhibitory receptor CD159a [29].

Nanosilver possesses anti-inflammatory properties in both animal models and in the clinic. Thus, AgNP inhibit the expression of proinflammatory cytokines transforming growth factor-β and tumor necrosis factor-α. Nanosilver administration attenuates nasal symptoms of allergic rhinitis in mice and inhibits immunoglobulin IgE, IL-4, and IL-10. In clinical study, wound dressing containing AgNP promoted the healing of chronic leg ulcers due to antibacterial effect in the wound and by decreasing inflammatory response. Ability of the nanosilver to reduce cytokine release and matrix metalloproteinases, to decrease lymphocyte and mast cell infiltration, and to induce apoptosis in inflammatory cells may explain their anti-inflammatory mechanisms [35].

Innate immunity induction with AgNO_3_ and AgNP conjugated with the major blood proteins—albumin, fibrinogen, and immunoglobulins—was assayed by Unknown (X) Multi Analyte Profiling (xMAP) using the kit “17 plex” (BioRad) (Table 1).

The maximal production of all 17 studied biomarkers including T-helper (Th)1 cytokines: interferon (IFN) γ, tumour necrosis factor (TNF) α, interleukin (IL-1β), IL-12 (p70); Th2 cytokines: IL-2, -4, -5, -6, -7, -8, -10, -13; Th17: IL-17A and other inflammation biomarkers: granulocyte colony-stimulating factor (G-CSF), granulocyte macrophage colony stimulating factor (GM-CSF), monocyte chemotactic protein 1 (monocyte chemotactic and activating factor) (MCP-1 (MCAF)), and macrophage inflammatory protein 1β (MIP-1β) were registered during the first two days posttreatment of human larynx carcinoma HEp-2, oral epithelial carcinoma L41, and colorectal adenocarcinoma HT-29 cells.

In HEp-2 human cell line in 2 days post-treatment with 5 µg/mL AgNO_3_ (for comparison, cytotoxic concentration for 50% cells (CC_50_) of AgNO_3_ for HEp-2 cells is 50 µg/mL), the significant upregulation was registered for IL-1β (slight increase of 1.38 times up to 0.08 pg/mL) and IL-8 (significant growth of 3.7 times up to 0.94 pg/mL), whereas steady production of TNF-α (growth of 1.15 times), G-CSF (up at 1.12), and MCP-1 (MCAP; increase of 1.17 times up to 0.39 pg/mL) was observed, whereas downregulation was observed for regulatory IL-10 (below the IL-10 production in control intact cells with less than 0.05 pg/mL and therefore undetectable in ELISA with detection limit 1 pg/mL) (Table 1). IL production is known to depend on the origin of human cells. Thus, IL-1β is mainly produced by macrophages and monocytes, but HEp-2 cell line is derived from larynx carcinoma cells and HT-29, from human colorectal adenocarcinoma cells. IL-1β is responsible for initiation and regulation of inflammation; stimulation of acute phase IL-2, -3, -6, and TNF-α; and for temperature growth and fever. Therefore, inflammation is hardly possible with a low level of IL-1β and stimulated cytokines. The only exception was IL-1β-induced upregulation of IL-8 (Table 1), which is associated with acute and chronic inflammation. Silver ions at a concentration lower than CC_50_ were not toxic for the human cell line and could not penetrate through membranes in cells. Therefore, the observed immunomodulation with AgNO_3_ was particularly modest (if at all).

Immunomodulation with nanoconjugates of AgNP covered with the major blood proteins—albumins, fibrinogen, or immunoglobulins—differed (Table 1). Significant increase of 2.11 times of IL-1β production (up to 0.12 pg/mL) and G-CSF (of 1.20 times) was detected in 2 days after treatment of HEp-2 cells with AgNP-hIgG simultaneously and similar to its growth after AgNO_3_ addition. However, the growth did not stimulate production of other cytokines (Table 1). Growth of MCP-1 (MCAP) secretion (up to 0.37 pg/mL) was caused by AgNP-fibrinogen. Inhibition of IL-8 and regulatory IL-10 production in the presence of all nanoconjugates with Ag core and protein shells resulted in slight changes (if any) of 17 biomarkers. It is worth noting that AgNP-BSA caused decreased production of all inflammation biomarkers studied (Table 1). However, the lowest concentrations below 50% of the corresponding values in control cells were found after treatment with AgNP-hIgG, perhaps due to reverse regulation of innate immune response with antibodies.

## 6. Conclusions

Limited stability of Ag nanostructures without stabilizing envelopes in biologically relevant media, cytotoxicity for both eukaryotic and bacterial cells, and negligible cellular uptake restrict their further implementation for combined antiviral and antibacterial therapy. However, spontaneous binding with the major blood proteins and anti-inflammatory properties with inhibition of cytokine production at the early stages after treatment may be helpful in prevention of the cytokine storm caused by RNA-containing viruses.

## Figures and Tables

**Figure 1 ijms-22-09928-f001:**
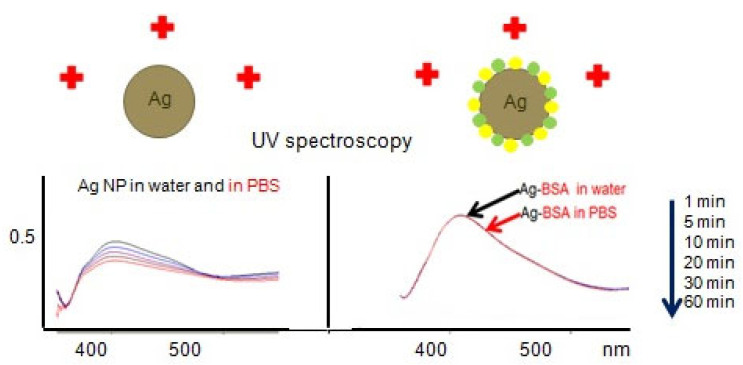
Comparison of stability of citrate coated AgNP (left panel) and the same AgNP conjugated with bovine serum albumin (BSA) (right panel) in water and phosphate buffer solution (PBS) by UV–VIS spectroscopy.

**Table 1 ijms-22-09928-t001:** Unknown (X) Multi Analyte Profiling (xMAP) results for 17 inflammation biomarkers in HEp-2 cells in 2 days post-treatment. Normalization was carried out as a ratio of mean fluorescence intensity (MFI) for wells after incubation with the nanosilver to MFI of control intact HEp-2 cells.

	Th1				Th2								Th17	Others			
Inflammation Biomarkers	IFNγ	TNFα	IL-1β	IL-12(p70)	IL-2	IL-4	IL-5	IL-6	IL-7	IL-8	IL-10	IL-13	IL-17A	G-CSF	GM-CSF	MCP-1 (MCAP)	MIP-1β
AgNO_3_	1.03	1.15 ↑	1.38 ↑	0.70	0.74	0.84	0.88	0.85	0.70	3.71 ↑	0.78	0.81	0.97	1.12 ↑	0.92	1.19 ↑	1.17 ↑
AgNP-BSA	0.76	0.72	0.70	0.70	0.82	0.74	0.70	0.75	0.50	0.64	0.78	0.69	0.76	0.78	0.79	0.85	0.76
AgNP-Fb	0.72	0.65	1.00	0.72	0.78	0.75	0.82	0.75	0.55	1.00	0.77	0.83	0.84	1.00	0.83	1.15 ↑	0.97
AgNP-IgG	0.68	0.51	2.11 ↑	0.54	0.48	0.43	0.60	0.41	0.44	0.34	0.54	0.50	0.97	1.20 ↑	0.83	0.76	0.66

## Data Availability

Data supporting reported results can be demonstrated on request.

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
