# Peer review of "Silver Nanostructures: Limited Sensitivity of Detection, Toxicity and Anti-Inflammation Effects"

_ijms, 2021, doi:10.3390/ijms22189928_

Round 1
Reviewer 1 Report
The author did not provide the track-changed document for this reviewer to see what new material has been added.
Some places the spacing between lines the paragraph changes, please make it uniform.
I noticed some additional references but author could add the following review on microwave-assisted synthesis (on line 62):
Microwave-Assisted Green Synthesis of Silver Nanostructures
Acc. Chem. Res., 44, 469-478 (2011).
With this change, it may be accepted for publication
Author Response
- The author did not provide the track-changed document for this reviewer to see what new material has been added.
Answer:
The original manuscript was essentially revised according to recommendations of 3 reviewers. Thus, original review included 17 references only, mainly recently published reviews devoted to nanosilver toxicity, antibacterial and antiviral properties without description of mechanisms of actions whereas the revised version was based on 35 published papers with mechanisms and our previously unpublished experimental observations. Therefore, new text significantly differed from the first version. It was hardly possible to mark all numerous changes. At the current stage of additional revision both track-changed document and final text are submitted for possible comparison.
- Some places the spacing between lines the paragraph changes, please make it uniform.
Answer:
I appreciate your attention, careful reading and hope to solve all the issues raised.
- I noticed some additional references but author could add the following review on microwave-assisted synthesis (on line 62):
Microwave-Assisted Green Synthesis of Silver Nanostructures
Acc. Chem. Res., 44, 469-478 (2011).
Answer:
The recommended reference was added in the re-revised manuscript.
Reviewer 2 Report
In manuscript entitled “Nanosilver in biomedicine: advantages and restrictions” Authors have presented resubmitted version of review. I still recommended major revision of work. First of all problem with structure of work, more example of utilization of Nanosilver, silver oxide NPs or AgCl NPs? in medicine. Why bio? medicine in not bio? Nanosilver in tile is misunderstanding.
What about formation mechanism in different system in context of its potential medical application ?
Minor concern
Authors stated: Trojan horse’, inductive and quantum-mechanical which may have a collaborative effect, e.g please detail.
AgNP can serve as Ag+ carriers with permanent release of Ag+ ions through the oxdative dissolution? Why ? what about termodynamic processes and its implications?
In my opinion manuscript required the major revision before final acceptance.
Author Response
- In manuscript entitled “Nanosilver in biomedicine: advantages and restrictions” Authors have presented resubmitted version of review. I still recommended major revision of work. First of all problem with structure of work, more example of utilization of Nanosilver, silver oxide NPs or AgCl NPs? in medicine. Why bio? medicine in not bio? Nanosilver in tile is misunderstanding.
Answer:
The title was changed to the following “Silver nanostructures: limited sensitivity of detection, toxicity and anti-inflammation effects”. Broad implementation of silver nanostructures was previously described with relevant references. Nevertheless, restrictions of the wide utilization are currently becoming evident and worth to discuss.
- What about formation mechanism in different system in context of its potential medical application?
Answer:
“Green synthesis” of Ag nanostructures by using various plant extracts, bacteria, biodegradable polymers, enzymes and even microwaves is gradually replacing harmful chemical synthesis and energy-consuming physical approaches of the nanosilver fabrication. These numerous reducing agents provide interactions with Ag+ ions, their recovery with subsequent nucleation and growth of Ag nanostructures. Due to high surface energy a few original Ag atoms and tiny nanoclusters tend to aggregate into nanoparticles. Microwave chemistry is based on the dipolar mechanism and the electrical conductor mechanism. For potential medical application AgNP should remain stable without leakage of toxic Ag+ ions capable to react with thiol, amino, phosphate and carboxyl groups of biopolymers. Protein shells of Ag nanostructures with the main blood proteins (immunoglobulins, fibrinogen and albumins) efficiently protect these extracellular nanoconjugates but cellular uptake by means of endocytosis results in proteolytic degradation of proteins in lysosomes and immediate intracellular Ag release (so called ‘Trojan horse’ effect).
Minor concern
Authors stated: Trojan horse’, inductive and quantum-mechanical which may have a collaborative effect, e.g please detail.
Answer:
Pryshchepa et al., 2020 suggested 3 mechanisms of nanosilver toxicity: ‘Trojan horse’, inductive and quantum-mechanical. All of them are described in details in the corresponding original paper and in the revised text. It is hardly possibly to separate them. Thus, cellular uptake of Ag nanostructures by endocytosis causes membrane damages (inductive mechanism), endosome formation and intracellular delivery of AgNP into lysosomes with ‘Trojan horse’ effect of Ag+ ions release simultaneously with electrostatic interaction with membrane proteins and phospholipids causing intracellular membrane degradation (inductive mechanism) along with electron-electron interactions with ROS formation and inhibition of cellular anti-oxidant enzymes (quantum-mechanical mode). Besides that, there are multiple points of interaction that hardly possible to distinguish. According to the recommendation, the evident statement concerning “collaborative effect” was removed from the revised abstract.
- AgNP can serve as Ag+ carriers with permanent release of Ag+ ions through the oxdative dissolution? Why ? what about termodynamic processes and its implications?
Answer:
The NP stability depends on the affinity of coating molecules to the particle surface, repulsion from neighboring molecules, loss of chain entropy upon adsorption, and also nonspecific dipole interactions between the macromolecule, the solvent, and the surface. Uncoated AgNP with metallic core and the surface shell - Ag2O are dissolved or aggregate in the presence of ions due to replacement of Ag+ by electrolyte ions, potential formation of insoluble AgCl, subsequent catalyzed oxidative corrosion of Ag and further dissolution of surface layer of Ag2O. AgNP indeed can serve as Ag+ carriers as well as damage surface and intracellular membranes during their cellular uptake by unspecific endocytosis.
- In my opinion manuscript required the major revision before final acceptance.
Answer:
All the recommendations were taken into consideration and the manuscript was additionally revised.
Round 2
Reviewer 2 Report
In my opinion revised manuscript can be accept in present form.
This manuscript is a resubmission of an earlier submission. The following is a list of the peer review reports and author responses from that submission.
Round 1
Reviewer 1 Report
This review is describing the effect of the surface chemistry (thiol, citrate ...etc.) of different types silver nanoparticles on their applications in several fields as antibacterial, antiviral, antifungal and anti-cancer agents. The stabilization of these nanoparticles in the medium depends on this surface chemistry, which directly affect their cytoxicity and host immune response. This review is divided into four parts: first, the protein corona formation on citrate-coated silver nanoparticles. Second part is the application of silver nanoparticles in immunodiagnostics. Third part is the cytotoxicity of silver ions and nanoparticles and finally, the inhibition of innate immunity with the nanosilver.
My opinion is that the review is not structured, the originality of this review compared to other reviews in same area is not clear. There is a huge lack of bibliographic references in this domain, for example:
- Ahamed, M., M. S. AlSalhi, et al. (2010). "Silver nanoparticle applications and human health." Clinica Chimica Acta 411(23-24): 1841-1848.
- Beer, C., R. Foldbjerg, et al. (2012). "Toxicity of silver nanoparticles-Nanoparticle or silver ion?" Toxicology Letters 208(3): 286-292.
- Chernousova, S. and M. Epple (2013). "Silver as Antibacterial Agent: Ion, Nanoparticle, and Metal." Angewandte Chemie-International Edition 52(6): 1636-1653.
- Pal, S., Y. K. Tak, et al. (2007). "Does the antibacterial activity of silver nanoparticles depend on the shape of the nanoparticle? A study of the gram-negative bacterium Escherichia coli." Applied and Environmental Microbiology 73(6): 1712-1720.
- Reidy, B., A. Haase, et al. (2013). "Mechanisms of Silver Nanoparticle Release, Transformation and Toxicity: A Critical Review of Current Knowledge and Recommendations for Future Studies and Applications." Materials 6(6): 2295-2350.
Other remarks: 2 figures are probably missing (cited in lines 85 and 100).
The Table cited at the end of the review is not clear and not easy to understand.
Reviewer 2 Report
Author has described antibacterial, antiviral, antifungal and anti-cancer properties of Ag-based nanomaterials possess anti-inflammatory, anti-angiogenesis and antiplatelet features with some comments on drug efficacy and the stability imparted by citrate coated Ag nanoparticles (NPs) in deionized water but not in the presence of ions due to replacement of Ag+ by electrolyte ions, potential formation of insoluble AgCl, subsequent catalyzed oxidative corrosion of Ag and further dissolution of surface layer of Ag2O. The above information and the protein shells protection accorded to AgNPs core from oxidation, dissolution, aggregation is well known. Author should have given some review articles for folks to follow besides citrate, such as:
Synthesis of Silver and Gold Nanoparticles Using Antioxidants from Blackberry, Blueberry, Pomegranate and Turmeric Extracts. ACS Sustain. Chem. & Eng., 2, 1717-1723 (2014). DOI: 10.1021/sc500237k.
Greener Techniques for the Synthesis of Silver Nanoparticles using Plant Extracts, Enzymes, Bacteria, Biodegradable Polymers and Microwaves: ACS Sustain. Chem. & Eng., 1, 703-712 (2013). DOI: 10.1021/sc4000362.
And
Silver-based Antibacterial Surfaces for Drinking Water Disinfection-An overview. Curr. Opin. Chem. Eng., 3, 25-29 (2014).
There is so much known on the subject as Ag nanoparticles have been most used among all others, yet author had only 17 references and none really of the authoritative reviews to benefit the reader.
This is a very limited review which covers only superficially some aspects and not quite comprehensive.
This reviewer cannot recommend this publication unless it is heavily revised with only handful of important topics.
Reviewer 3 Report
In manuscript entitled “Nanosilver in Biomedicine: Advantages and Restrictions” Author have application of AgNPs, a specially in various filed of biomedicine: like immunodiagnostics, .
This paper should be revised according to the following comments.
- Please precise in abstract and in whole text type of binding of what silver ions, silver oxide, chloride or AgNPs to active functional group
- Please provide additional discussion about:
- different type of action mechanism, except of Trojan horse, a specially induced model see Adv Colloid Interface Sci. 2020 Oct;284:102246)
- formation mechanism of AgNPs
- what about problem with contamination of AgNPs such as AgCl and their inculace on cytotoxicity?
- What about imaging techniques with AgNPs used in biomedicine ?
In my opinion manuscript required the major revision before final acceptance.